# Structural Characterization and Spatial Mapping of Tetrodotoxins in Australian Polyclads

**DOI:** 10.3390/md20120788

**Published:** 2022-12-19

**Authors:** Justin M. McNab, Matthew T. Briggs, Jane E. Williamson, Peter Hoffmann, Jorge Rodriguez, Peter Karuso

**Affiliations:** 1School of Natural Sciences, Macquarie University, Sydney, NSW 2109, Australia; 2Department of Applied Biosciences, Macquarie University, Sydney, NSW 2109, Australia; 3Clinical & Health Sciences, University of South Australia, Adelaide, SA 5000, Australia; 4School of Biotechnology & Biomolecular Science, University of NSW, Sydney, NSW 2052, Australia

**Keywords:** tetrodotoxin, Polycladida, HILIC-HRMS, flatworm, MALDI-MSI

## Abstract

Tetrodotoxin (TTX) is a potent marine neurotoxin that occurs in several Australian phyla, including pufferfish, toadfish, gobies, and the blue-ringed octopus. These animals are partially immune, and TTX is known to bioaccumulate and subject to trophic transfer. As such, it could be more ubiquitously distributed in animals than is currently known. Flatworms of the order Polycladida are commonly occurring invertebrates in intertidal ecosystems and are especially diverse in Australian waters. While TTX has been identified in polyclads from Japan and New Zealand, Australian species have yet to be tested. In this study, several eastern Australian polyclad flatworm species from the suborders Cotylea and Acotylea were tested for TTX and analogs by HILIC-HRMS to understand the distribution of this toxin within these suborders. Herein, we report the detection of TTX and some known analogs in polyclad species, one of which is a pest to shellfish aquaculture. We also report, for the first time, the application of MALDI mass spectrometry imaging utilized to map TTX spatially within the intestinal system of polyclads. The identification of TTX and its analogs in Australian flatworms illustrates a broader range of toxic flatworms and highlights that analogs are important to consider when studying the distributions of toxins in animals.

## 1. Introduction

Marine ecosystems are comprised of mosaics of natural products that are utilized by animals to interact with other members of the biological communities they inhabit. Defensive or offensive natural products utilized by marine organisms are often synthesized from their food, acquired from symbionts, or endogenously produced [1] and can impact the demography of other organisms within their communities [2,3]. As simple, soft-bodied organisms that are often limited in mobility, many marine invertebrates contain a suite of natural products, with sponges, ascidians, and nudibranchs all heavily reliant on chemical defenses [4,5,6]. In the broad range of natural products used by invertebrates to defend against predation, a particularly notable and effective group are neurotoxins, molecules that affect the excitability of nerve cells and cause paralysis [7,8]. One prominent group of natural product neurotoxins, sodium channel blockers, function by binding to specific sites of positive-gated sodium channels, which in turn inhibit a neurotransmitter that is crucial for muscle contraction in animals [8]. Well-studied sodium-blocking neurotoxins in the marine environment include tetrodotoxin, saxitoxin, ciguatoxin, brevetoxin, and conotoxins [7,8,9]. Despite their extreme toxicity to animals, several marine neurotoxins are utilized by a range of taxa as they provide defensive and offensive benefits if autotoxicity can be avoided [10].

Tetrodotoxin (TTX, Figure 1, **1**) is a non-proteinaceous neurotoxin of mixed biogenesis from arginine, isopentenylpyrophosphate, and an apiose-type C5 sugar [11]. Tetrodotoxin is a guanidinium-based tricyclic neurotoxin that binds to the SS2 region of voltage-gated sodium channels in neural and intramuscular cells. Specifically, the hydroxyls at C-6, C-9, C-10, and C-11 are essential for the high binding affinity that result in forcing these channels closed, resulting in an inability to depolarize, which causes paralysis in organisms. The molecule binds to the SS2 region of repeat 1 of positive-gated sodium channels in neural cells [12]. Tetrodotoxin is utilized by a range of phyla, including vertebrates and invertebrates, typically either in a defensive context (e.g., poison in pufferfish [13]) or an offensive context (e.g., venom in octopus [14]). However, it can also act as a semiochemical for some pufferfish [15,16]. There is also evidence of TTX in invertebrate communities in a diversity of phyla, including echinoderms, decapods, platyhelminthes, gastropods, and mollusks that have been discovered from a range of locations globally, and it is expected that the organisms that contain TTX utilize it [11,17,18,19]. Due to this pervasive distribution, this toxin is considered to affect the ecology of environments it occurs in, as organisms that can utilize it flourish whilst excluding organisms that are not tolerant [20]. It is for this reason that TTX is considered a ‘keystone chemical’ that is important in ecological studies [21].

Marine platyhelminths of the order Polycladida are tiny but voracious mesopredators in littoral and benthic communities, and some species have recently been shown to contain TTX and various analogs in their ovaries, eggs, and pharynges [22]. Due to the distribution of toxins in these areas, species that contain TTX are thought to utilize this molecule to assist in predation [23,24]. Polyclads from Asia and New Zealand from the Acotylean suborder have been shown to contain TTX, and the predominant genera with toxic species are *Planocera* [23,25]. If this toxin is confined within a single genus, it suggests a weak phylogenetic relationship with a molecule that would be greatly beneficial to these animals, although only a few genera of Acotylea have been tested for TTX [25]. Australia has a large and widespread diversity of polyclad flatworms from both the suborders of Acotylea and Cotylea, suggesting they are quite successful in Australia’s waters [26,27,28,29].

Qualitative research concerning TTX is common as this toxin occurs within food items, particularly within aquaculture [30]. As such, a variety of methods have been tested to investigate this toxin and its analogs. These include bioassays, histological assays, functional bioassays and ELISA, and various chromatography techniques such as GC-MS, HPLC-FLD with derivatization, and LC-MS/MS [10]. Historically, biological assays have been commonplace when testing for TTX, with particular emphasis on mouse bioassays, to a scale that justified the adoption of the MU (mouse unit) when trying to quantify toxin concentrations from samples [31,32]. Furthermore, several histological methods allowed researchers to identify the toxin location within the animals, enabling the identification of storage locations of this toxin [32,33]. The limitations of this indirect method are in differentiating other neurotoxins and analogs from TTX. Mouse units appear to have a disparity with TTX concentration, reported by different authors, possibly because of interference by different compounds and differences between the mice used for the assay.

In contrast, direct methods such as LC-MS/MS can be quantified and used to detect TTX analogs but result in loss of spatial information. In order to overcome this limitation, techniques such as matrix-assisted laser desorption/ionization mass spectrometry imaging (MALDI-MSI) could be used to spatially map and visualize analyte distribution across a tissue section [34,35,36]. In the context of marine natural products, this technique has been utilized to illustrate the distribution of metabolites in cone snails (*Conus geographus* and *Conus marmoreus*) [37] and a Muricid snail (*Dicathais orbita*) [34,38] for example. The MALDI-MSI investigation of *C. geographus* revealed that two distinct venom types are released depending on whether the cone snail is interacting with prey or a potential predator, with each type localized to separate secretory glands [37]. This study demonstrated the complexity of conotoxins produced by these animals and illustrated the value of employing MALDI-MSI in natural products research. To the best of our knowledge, this is the first report on the application of MALDI-MSI to any flatworm.

Our aim was to first ascertain whether TTX, or analogs, occur in polyclads from temperate Eastern Australian waters using LC-MS/MS and, if so, secondly, where the toxins are located within flatworms using MALDI-MS. The advantage of LC-MS/MS is that it can detect trace levels of metabolites, while MALDI-MS provides spatial information on the cellular location of the metabolites. Together, this information will help clarify the geographical range of TTX in flatworms and in the marine communities of this area more generally and help elucidate the ecological and evolutionary roles of neurotoxins in marine systems.

## 2. Results

A total of 32 flatworms were collected from several locations in NSW (Appendix A). Four of these specimens were used for MALDI-MSI, and the remaining 28 were assessed for the presence of TTX and its analogs using hydrophilic interaction liquid chromatography-mass spectrometry (HILIC-HRMS) and high-resolution parallel reaction monitoring (PRM) of daughter ions (Appendix A). Additionally, of these 28 specimens tested for HILIC-HRMS, 19 were subjected to molecular systematics to confirm the initial species identification. Of the species tested, polyclads from both Acotylea and Cotylea were screened for TTX and its analogs, with the results summarized in Table 1 (raw data in the supplementary material). Flatworm specimens used in HILIC-HRMS were identified as the genera of *Echinoplana* (*n* = 4), *Stylochus* (*n* = 12), *Notoplana* (*n* = 1), *Cycloporus* (*n* = 1), *Pseudoceros* (*n* = 5), *Eurylepta* (*n* = 1), *Thysanozoon* (*n* = 2), *Cestoplana* (*n* = 1) and *Tripylocelis* (*n* = 1) (Appendix A). Freeze-dried polyclads used for MADLI-MSI included *Stylochus* sp. 1 (*n* = 2) and *Notoplana* cf *longiducta* (*n* = 1).

### 2.1. Separation and Identification of TTX and Its Analogues Using HILIC-HRMS

To validate the HILIC-HRMS method, a TTX (**1**) standard (ABCAM) was initially run to assess separation and detection [39]. It was determined that the TTX compound eluted at 18.1 min (RSD 6.9%) (Figure 1A). TTX parallel reaction monitoring (PRM) product ions were observed at *m*/*z* 302.1, 284.1, 256.1, 178.1, 162.1, 146.1 (Figure 1A). However, as the signal obtained from individual flatworm extracts was 3300-fold lower, only the 178.1 and 162.1 PRM products were observable, with 0.15 and 0.06 Δmmu errors, respectively, confirming the presence of TTX in *Stylochus* cf *mcgrathi* (Figure 1B). PRM ions related to the other TTX analogs were taken from the literature [10,39], and when observed at a similar elution time, the analog was considered present in the sample (Appendix A). The results from each mass channel scanned by HILIC-HRMS are displayed for each sample in Appendix A.

A compound that could be TTX 11-carboxylic acid (**6**) was detected by scanning for a mass of 334.0881, eluting at about 7.5 min. There is no PRM data published for this compound, but prominent daughter ions at 197.09, 161.07, 149.07, 137.07, and 110.06 were observed (Appendix A, pages 74–80). Both analogs **3** and **4** also contain the 161.07 daughter ion that corresponds to C_9_H_9_N_2_O, and 110.06 was observed in analog **2** (C_6_H_8_NO). These data suggest this compound is a TTX analog, the only compound in the literature with this formula is **6** [10]. While this is a tentative assignment based on circumstantial evidence, it is biogenetically a simple oxidation of **1** or **5**. 

### 2.2. Optimization of Flatworm Sample Preparation for TTX MALDI-MSI

In order to spatially map TTX and its analogs across flatworm specimens, each specimen was fixed to ITO-coated slides. Initially, flatworms were fixed by washing them in aqueous ethanol (90% *v*/*v*) (30 min), ethanol (30 min), and xylene (30 min) to dehydrate the specimen. However, profiling using MALDI-TOF/TOF MS resulted in no detection of TTX or any of the known analogs (Figure 2). It is likely that the expected compounds had been washed out of the sample during the fixing/dehydration steps. In order to overcome this limitation, flatworms were freeze-dried, resulting in the detection of only the most common metabolite, 11-deoxyTTX (**2**) (Figure 2, Table 1). The only other ion observed was 332. Going back to the HILIC-HRMS (PRM acquisition mode) data, we were able to detect 332.19278 in all specimens of *Stylochus cf mcgrathi* (Appendix A, pages 81–88), which corresponded to a molecular formula of C_13_H_26_N_5_O_5_ (Δmmu = −0.07) that is an unknown metabolite not related to TTX.

### 2.3. MALDI-MSI of Freeze Dried Stylochus *sp*. 1 Flatworms

11-DeoxyTTX (**2**) was found to be localized in the rostral intestinal region of two replicates of *Stylochus* sp. 1. The unknown analyte at *m*/*z* 332.3 colocalized with **2** in both specimens (Figure 3), suggesting it might be a counterion. 

## 3. Discussion

Most flatworms collected from intertidal areas from coastal NSW possessed TTX analogs, but only *Stylochus* cf *mcgrathi* collected from Cronulla shellfish farm was found to contain TTX (**1,**
Table 1), which was identified with the fragmentation ion 162.067 that corresponded to C_8_H_8_N_3_O (Δmmu = 0.06, Appendix A pages 3–8). The similar retention times and chemical structure of the PRM ions indicate that these samples contained TTX despite not all PRM fragmentation ions being present. Flatworms used in this study likely had small concentrations of TTX, and as such, the remaining ions were below the detection limit of the mass spectrometer. The most found metabolite was 11-norTTX-(6*S*)-ol (**3**), which was found with a fragmentation ion *m*/*z* 162.075, which most closely corresponds with C_9_H_10_N_2_O (Δmmu = −1.27, Appendix A pages 18–41), this was detected in 86% of the samples. The rarest analyte was 4a-anhydro-5,6,11trideoxyTTX (**9**) with fragmentation ion 162.067 corresponding to C_6_H_12_NO_4_ (Δmmu = 0.53, Appendix A pages 72–73), which were detected in only two specimens. Other common analytes were 11-deoxyTTX (**2**), which was identified with daughter ion 176.07 that corresponded to C_10_H_10_NO_2_ (Δmmu = −1.83, Appendix A pages 9–17) and 6,11-dideoxyTTX (**4**) that was identified with 162.07 that corresponded to C_6_H_12_NO_4_ (Δmmu = −0.48, Appendix A pages 42–57). Unknown compound #1 (Table 1) is potentially 1-hydroxy-4,4a-anhydro-8-*epi*-5,6,11-trideoxyTTX (**10**) and was identified in several flatworm samples. Previously 8-*epi*-TTX analogs had only been identified from newts. However, a recent study potentially isolated two 8-*epi*-analogs from a *Cephalothrix* species collected from the Sea of Japan [40]. In this context, as neither this report nor our study had analytical standards for this compound, this analog could also be 4,9-anhydroTTX or 4,4a-anhydrodideoxyTTX, which have the same exact mass and more closely related to the other compounds isolated. Here, unknown compound #1 was isolated with the fragment ions 162.07 (C_8_H_8_N_3_O, Δmmu = −0.14) and 110.07 (C_5_H_8_N_3_ Δmmu = −0.04), which were also reported for compound **10** (Appendix A pages 58–71), but as these are common ions in tetrodotoxin analogs, this evidence also does not confirm the identity of this compound as **10**. 

These analytes were putatively observed by PRM filtering to reduce background noise and other compounds and with fragment ions that have been previously described and reported in the literature [10]. This, along with the relatively small difference in Δmmu between the hypothetical PRM ion mass and observed PRM ion masses and similar retention time of samples, support the assessment that these samples contain the analytes shown in Table 1. 

There were some phylogenetic trends, such as *Stylochus* cf *mcgrathi* always contained **3** and **8** and *Echinoplana* cf *celerrima* always contained **3** and **4**. *Stylochus* species did not all contain TTX analogs. However, and *Stylochus* sp 3 and sp 4 were the only samples do not to contain any analytes. The rest of the flatworm samples were quite variable in their analyte content suggesting geographic variations that might be related to diet. 

11-DeoxyTTX (**2**) was identified as the most concentrated analog and localized in the intestinal region in two specimens of *Stylochus* sp. 1 (Figure 3). Standards of the various analogs of tetrodotoxin are not available, but HRMS and PRM data were used to positively identify the known metabolites. These data corroborate previous results on the general location of these toxins [22]. However, previous results suggested the ovaries, eggs, and pharynges contained TTX, but our results show localization only in the intestinal region (Figure 3).

HILIC-HRMS data also putatively identified unknown compound #2 (Table 1) as TTX-11 carboxylic acid (**6**). This is a known compound but not previously seen as a natural product [41]. Evidence for identification of **6** is a close match for the HRMS (C_11_H_16_N_3_O_9_, Δmmu = −0.01) and daughter ions at 110.0613, 137.0795 that was also observed for compound **2**. In addition, a daughter ion at 161.07091 was also found in compounds **3** and **4**, suggesting that this compound is a TTX analog (Appendix A, pages 74–80). There are, however, no published data on the precise fragmentation pattern of this compound, and identification was based on peaks that were retained around 7.55 min from filtered PRM scans at *m*/*z* 334.0881 which is the protonated form of the TTX-11 carboxylic acid [40]. TTX-11 carboxylic acid (**6**) is a synthetic compound derived from TTX by oxidation and is known to be biologically inactive [41]. The distribution (Table 1) does not show a particular pattern, together with the known inactivity of this compound, suggesting that it may be an artifact of extraction, or a detoxification product produced by the flatworm. 

Determining the importance of TTX analogs within polyclads is difficult, as only a few other studies have previously attempted this [22,23,24]. Within the species, *Planocera* sp. and *Stylochus orientalis*, a range of analogs were isolated (Figure 1) [23,42]. Most recently, in *Planocera* sp., the first occurrence of 6,11-dideoxyTTX (**4**) was found among eight other analogs, and this led to the proposition that these molecules could be associated with the biosynthesis of TTX in marine animals [24]. Another possibility is that some of the compounds may be degradation products or metabolically elaborated, as has been shown in nudibranchs [4]. Results presented here show that with the exclusion of 11-oxoTTX (**5**), the same analogs occur in Australian polyclads. However, variation in toxin content between individuals was identified. The observed differences in species sampled could be due to differences in diet, loss of detection due to decreases in signal intensity, potential seasonal changes, or collection site of the animals. Polyclads were collected from the months of January –August 2020, and bacterial concentrations in these areas, for example, were not monitored and could contribute to variation in results (Table 1). The co-localization of the *m*/*z* 332.19 metabolites with **2** (Figure 3) suggests that this compound could be a counterion. The exact mass (Appendix A pages 81–88) fits very well for C_13_H_26_N_5_O_5_ (Δmmu = −0.07), which could be Lys-Ala-Asp-NH_2_ for example.

The results presented here are an example of another phylogenetic link between polyclads and TTX, as previously, the toxin had only been isolated from flatworms of the *Planocera* genus [25]. These HILIC-HRMS results show that TTX is positively identified from several individuals from *Stylochus cf mcgrathi* from both oyster and mussel farms. Both genera belong to the family Stylochidae and thus might share similar ecological niches, which would require TTX. This data confirms the assumptions made by several other authors that flatworms that predate on shellfish contain a type of neurotoxin that could assist in prey subjugation [43,44,45,46]. As Planocerid flatworms utilize TTX (and analogs) to broaden their hunting options to a larger diversity of prey [23], the Stylochid flatworms that the toxin was identified from here could be utilizing their TTX for highly nutritious food that is more difficult to obtain, and this could explain how they are so successful as a pest species.

Multiple analogs of TTX were isolated from both Cotylean and Acotylean flatworms. These species are likely to have varied diets, and the presence of analogs is intriguing as it could suggest that this is either related to specific locations, a common food being shared by all species and large distribution of TTX in benthic environments, or a symbiotic bacterium producing TTX in these polyclads. It is unknown if the combination of analogs found in the tested individuals is connected to a location, as the sample size of flatworms was too small to derive a meaningful trend. Of the observed analogs, 11-norTTX-6-(S)-ol (**3**), 6,11-dideoxyTTX (**4**), and unknown compound #1 were most commonly occurring in flatworm extracts, through individuals from both Cotylea and Acotylea. Tetrodotoxin, however, was only found at one location in *Stylochus* cf *mcgrathi,* and this could be an effect of the location, or the previous history of the individuals tested, such as the food these flatworms were eating. Previously, analogs have been described in *Planocera* sp. by Ritson-Williams, Yotsu-Yamashita, and Paul [23] and Yotsu-Yamashita, Abe, Kudo, Ritson-Williams, Paul, Konoki, Cho, Adachi, Imazu, Nishikawa, and Isobe [24], where the authors isolated **1**, **2**, **3**, **4**, **5**, **7**, **8** as well as the additional analogs not found here, such as 4-*epi*-TTX and 6,11-dideoxyTTX. Species tested in this study corroborated some of these results, as a similar range of analogs were observed in many tested species of polyclad. Although no individuals were reported to contain 11-oxoTTX (**5**), 4,9-anhydroTTX (**8**), 4-*epi*-TTX, or 6,11-dideoxyTTX, these similarities give support to the results presented in Table 1. In the paper by Ritson-Williams, Yotsu-Yamashita, and Paul [23], *Planocera* sp. was described as containing a broad diet, comprising several gastropods, bivalves, and another polyclad. The diet of most polyclad species is largely unknown, with only pest species being reported [47,48], and the breadth of these reported diets is likely to be underrepresented. As such, the ability to identify common prey items between species tested in this study is limited and beyond the scope of this research.

## 4. Materials and Methods

### 4.1. Flatworm Specimen Collection

Flatworms were collected by hand while walking at low tide or snorkeling at high tide from intertidal pebble beaches around NSW, Australia, in 2020 (Appendix A). Beaches included those of varying wave exposure and orientation and were chosen because of their representativeness of this habitat type in the region. Within each beach, a range of habitats was searched for flatworms, including under rocks of varying sizes, in algal beds, and from dead shellfish. Once an individual was located, it was gently removed with a fine paintbrush and individually held in fresh seawater while transported to the laboratory. The seawater was changed regularly to ensure individuals were kept cool and alive during transport. In the laboratory, individuals were placed in a Petri dish (145 mm diameter) with seawater, visually identified to genus, and photographed for verification of identification. A small piece of tissue (<1 mg) was removed from the lateral margin approximately halfway down the length of the animal, placed in ethanol, and stored at 4 °C for subsequent genetic sequencing. All equipment was cleaned thoroughly between individuals with ethanol and air-dried to avoid any cross-contamination. Individuals were then used for either hydrophilic interaction liquid chromatography-mass spectrometry (HILIC-HRMS) or matrix-assisted laser desorption/ionization mass spectrometry imaging (MALDI-MSI).

### 4.2. Tetrodotoxin Analogue Separation and Analysis from Flatworms Using Hydrophilic Interaction Liquid Chromatography Mass Spectrometry

To assess whether flatworms in this region contained TTX, solid phase extraction of different species was done following Salvitti, Wood, Winsor, and Cary [19]. First, each individual was manually homogenized with 100 µL of 0.01% *v*/*v* acetic acid in a glass Dounce blender. Next, samples were diluted with 1 mL of 0.01% *v*/*v* acetic acid in analytical-grade methanol. Extracts were then kept at −20 °C for 60 min and centrifuged at 3500 rpm for 10 min using a Beckman GS-15 centrifuge. A 400 µL aliquot was taken and concentrated to 40 µL by evaporating methanol using a stream of nitrogen gas. An aliquot of 20 µL was taken from the extract and diluted with 30 µL of methanol, then purified through a short column of C18 silica that was manually packed into a 200 µL pipette tip with a filter paper plug at the top and bottom of the tip. Before use, tips were washed with 40 µL of acetonitrile and 40 µL of MilliQ water for equilibration. Flatworm aliquots were then loaded into the tips and eluted with 35 µL of acidic aqueous methanol (20% *v*/*v*, 0.1% *v*/*v* acetic acid). Elutants were then spun on a benchtop centrifuge (Micro ONE MA-1, Tomy Tech, San Diego, CA USA), and 30 µL was removed for HILIC-HRMS analysis. 

Before samples were tested by HILIC-HRMS, a TTX standard (ABCAM, Cat. No: ab120054) was analyzed on a SeQuant ZIC-HILIC column (150 × 2.1 mm) with guard (20 × 2.1 mm) as per Bane, Brosnan, Barnes, Lehane, and Furey [39]. This was done on a Q Exactive Plus hybrid Quadrupole-Orbitrap mass spectrometer (Thermofisher Scientific, Waltham, MA, USA) with a Vanquish Horizon Ultra High-Performance Liquid Chromatography (UHPLC) system with heated electrospray ionization (HESI) source set in positive polarity mode for all testing. HESI parameters were; capillary temp: 320 °C, sheath gas temp: 50 units, and auxiliary gas flow: 10 units, ion spray voltage 3500 mV. HILIC-HRMS was performed at 40 °C with a flow rate of 300 µL min^−1^ with an initial backpressure of 43 bar. Xcalibur (v 4.0, Thermo Scientific) software was used to quantify TTX and its analogs. The TTX standard showed good peak shape and enabled the testing of flatworm samples.

HILIC-HRMS was performed with mobile phase conditions as follows: A: 0.01% acetic acid/5 mM ammonium acetate in Milli-Q water, B: 0.01% acetic acid/5 mM ammonium acetate in 90% acetonitrile. Mobile phase gradient profile involved: (1) 0 min, 3% A, (2) 15 min, 50% A, (3) 25 min 50% A, (4) 25.01 min, 3% A, (5) 35 min, 3% A, with a total run time of 35 min [49]. Flatworm samples were run in sequences interspersed with blank solvent samples (20% *v*/*v* methanol, 0.1% *v*/*v* acetic acid in MilliQ water) to avoid cross-contamination and clean between injections. Injection volumes of samples into the mass spectrometer were 5 µL. In order to ensure that MS was optimally detecting the analyte, a TTX standard was run at the beginning of testing, after every 5 samples, and at the end of testing. Full mass spectrum scanning was performed in the mass range *m*/*z* 50–750. Parallel reaction monitoring (PRM) was also performed to confirm the presence of TTX (320.1 > 162.1), as described in [39]. Additional PRM channels were assessed from [10] to quantify TTX analogs and are provided in Appendix A. If the product ions were observed at similar elution times, the analog was considered present. 

### 4.3. Spatial Mapping of TTX across Flatworms Using Matrix-Assisted Laser Desorption/Ionization Mass Spectrometry Imaging

Individual flatworms were coaxed onto indium tin oxide (ITO 8–12 Ω, 25 × 25 mm) microscope slides (Sigma Aldrich, Castle Hill, Australia), with a fine paintbrush, and then the slides were then placed onto frozen seawater to relax the animals into a completely flat position. Polyclads were then placed in a −20 °C freezer for 10 min and then transferred to −80 °C for a further 10 min. The individual was then put into a desiccator to dehydrate overnight. In order to estimate the sample thickness and predict laser penetration depth, several flatworms were measured using previously paraffin-embedded samples. This involved sagittal sectioning of paraffin-embedded flatworms using a Leica rotary microtome. Once the pharynx of the flatworm was visible, sections were put onto glass slides and photographed using an Olympus SZX 16 stereomicroscope, Notting Hill, Australia, next to a measurement standard. These images were then measured using ImageJ (version 1.52a) and averaged for thickness. Sections that were measured had an average thickness of 183 µm. 

### 4.4. Calibrant and Matrix Deposition

Firstly, red phosphorous (Sigma-Aldrich, Castle Hill, Australia) was mixed with methanol (1 mg/mL) and manually spotted (1 μL) onto each flatworm to externally calibrate the UltrafleXtreme MALDI-TOF/TOF MS instrument (Bruker Daltonics, Bremen, Germany). Slides were marked at the edges with water-based white out, and then scanned at 4800 dpi on a CanoScan 5600F (Canon Australia, Macquarie Park, Australia), scanner to teach the instrument the spatial orientation of the slide in relation to the laser. Subsequently, α-cyano-4-hydroxycinnamic acid (CHCA; 7 mg/mL in 50% acetonitrile, 0.2% *v*/*v* Trifluoroacetic acid) was deposited onto each slide using an iMatrixSpray, Subingen, Switzerland, instrument, and the following instrument-specific settings: 6 mm height with 1 mm line distance. Instrument speed was at 160 mm/s. The matrix cover was 1 μL/cm^2^ (9.8 units/cm^2^) density, 30 cycles, 15 s delay, and 80  ×  30 mm dimensions.

### 4.5. MALDI-TOF/TOF MS Data Acquisition

MS data were acquired using the ultrafleXtreme MALDI-TOF/TOF MS instrument controlled by FlexControl (version 3.4, Bruker Daltonics, Preston, Australia) and FlexImaging (version 4.0, Bruker Daltonics, Preston, Australia) in positive reflectron mode. Instrument-specific settings were as follows: *m*/*z* 50–1000 range, 2 kHz laser repetition rate, 22% laser power, and 2713V detector gain. A total of 2000 shots were acquired at each position using the Smartbeam II laser. A laser diameter (2_small) was used with a random walk within a 50 μm raster width. The MALDI-TOF/TOF MS instrument was externally calibrated using a quadratic fit prior to acquisition using the red phosphorous spotted prior to MALDI-MSI.

### 4.6. MALDI-TOF/TOF MS Data Analysis

MALDI-MSI data were analyzed and visualized using the SCiLS Lab software package (version 2016b, SCiLS, Bruker Daltonics). Raw data were loaded and pre-processed by the default settings for baseline subtraction (TopHat) and normalized to total ion count (TIC). Ion intensity maps were generated by manually selecting *m*/*z* values within a ± 0.3 Da window and confirmed by HILIC-HRMS. These ion intensity maps were weakly denoised with automatic hotspot removal. 

### 4.7. Flatworm Specimen Identification

Traditionally, flatworms are identified by sectioning the animal and assessing the internal morphology in conjunction with external morphology [43]. As different species of flatworm can share many externally similar characteristics and assessing the internal morphology was impractical because whole animals were used for all extractions, we verified species identification using molecular systematics, sequencing the mitochondrial COI (COX1) gene. DNA samples were collected using the Bioline Isolate II genomic DNA extraction kit according to the manufacturer’s instructions. The resulting PCR’s were conducted using Taq DNA polymerase (Mastermix) as directed in the manufacturer’s instructions (Invitrogen, Carlsbad, CA, USA). COI sequences were amplified with Acotylea_COI_F (5′-ACTTTATTCTACTAATCATAAGGATATAGG-3′) forward primer and Acotylea_COI_R (5′-CTTTCCTCTATAAAATGTTACTATTTGAGA-3′), reverse primer from Oya and Kajihara [50]. DNA of each species was compared to data from Rodríguez’s collections (unpublished data), or sequences found on the NCBI GenBank database. NCBI accession numbers for all tested polyclads are provided in Appendix A.

## 5. Conclusions

Here, we demonstrate that the neurotoxin, tetrodotoxin, and several analogs, occur within Australian polyclad flatworms from both Acotylea and Cotylea and putatively identified TTX-11 carboxylic acid (**6**) as a natural product and the first 8-*epi*-analogue, 1-hydroxy-4,4a-anhydro-8-*epi*-5,6,11-trideoxyTTX (**10**) from a flatworm but these results remain to be confirmed. We have also demonstrated, for the first time, that MALDI-MSI can be used to spatially map TTX across flatworms to complement LC-MS/MS data. This work provides a better understanding of TTX in polyclads and broadens the known range of these species to Australia, which often shares similar species and ecology with New Zealand. The presence of TTX in Stylochid flatworms also illustrates that this toxin occurs in a genus beyond *Planocera*, which broadens the taxonomic distribution of this toxin within Polycladida. It has been postulated that Stylochid flatworms contain TTX to assist in the predation of shellfish, as species taken from commercial farms possessed TTX and were previously known to be capable hunters of difficult prey like oysters.

More information is required about the source of TTX in marine systems. Neurotoxins, such as TTX, are well documented in higher trophic levels [13,51], but the concentrations in which they occur in benthic ecosystems and marine sediments are often unclear. Previous studies have isolated TTX-producing bacteria from deep-sea sediments and sublittoral zones, and this could possibly allow benthic species to sequester microflora for TTX utilization [52,53,54]. TTX-producing microbes have yet to be isolated from Australian sediments, and this could pose as the source of the toxin for polyclad flatworms.

## Data Availability

All mass spectrometry data is available in the Appendix A.

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
