# Peer review of "Structural Characterization and Spatial Mapping of Tetrodotoxins in Australian Polyclads"

_marinedrugs, 2022, doi:10.3390/md20120788_

Round 1

Reviewer 1 Report

This is an interesting manuscript that reports identification of tetrodotoxin and its analogues in several eastern Australian polyclad flatworm species using HILIC-MS and MALDI mass spectrometry imaging. I think that this manuscript is worth publishing in this Journal. However, I found one major point that should be revised.  

The authors identified the TTX analogue detected at [M+H]+ m/z 270.1085 as 1-hydroxy-4,4a-anhydro-8-epi-5,6,11-trideoxyTTX. However, 8-epi type analogues have been reported only in newts, not in marine animals. This ion may be identified as 4,9-anhydro-6,11-dideoxyTTX if 6,11-dideoxyTTX is detected in some species of flatworm as shown in Table 1. 

Therefore, I think that all description of 1-hydroxy-4,4a-anhydro-8-epi-5,6,11-trideoxyTTX should be corrected to 4,9-anhydro-6,11-dideoxyTTX in Figure 1, Table 1, and text, in whole manuscript.

Reviewer 2 Report

The manuscript requires a detailed revision, to improve the chemical description of the analytical methodologies, and to provide a more analytical discussion of the results. The work results from the application of a powerful chemical analysis tool, therefore it requires a rigorous discussion of all the chemical information, both at the chromatographic level and structural elucidation. The manuscript needs rigorous review. The title is not in harmony with the content of the work, there are more structural elucidation data than mapping image, the authors must review the document and reorganize. There is no harmony in the use of technique acronyms. A revision is required in the way concentrations of solutions are expressed. Material details and procedures on the sample preparation part are required, especially what refers to SPE.

Author Response

see the PDF file from the referee and our response attached

Reviewer 3 Report

This is a well-designed study with interesting findings TTXs occurrence in polyclads sampled from Australian waters. There are only few points needing some improvement in order to make the manuscript more reader friendly.

General remarks:

A. Throughout the text: when referring to other works using author names, use only the first author and then “et al.” – no need to write all author names (multiple instances in the text).

B. It would be beneficial for the manuscript (and the readers) to add some quantitative data on the presence of TTXs (even if quantified against the parent TTX compound).

Specific remarks:

Abstract:

- Page 1, line 20: “which is a pest…”: correct to “which are a pest”.

2. Results

- Page 4, lines 132-136 (Table 1): 

   * 2nd row: Could this be “Stylochus sp. 2”?

   * 12th, 13th & 16th row, 11-deoxyTTX column: Please check that these retention times are correct, they seem to be very different from the other cases. If it is not a mistake, please explain why this discrepancy could be present.

4. Materials and Methods 

- Page 9, line 260: Please close the parenthesis after S1.

- Page 9, line 279: “manually homogenizing each individual…”: what was the sample quantity of each individual that could be homogenized with only 100μl solution? Was it the whole one?

- Page 9, line 283: Was 400μL the total quantity of the supernatant or an aliquot of 400μL was taken? It is not clear from the text.

- Page 10, line 336: Please explain what the abbreviation “TFA” stands for.

Round 2

Reviewer 1 Report

I still cannot agree to the identification of 1-hydroxy-4,4a-anhydro-8-epi-5,6,11-trideoxyTTX. The authors of the present manuscript, and also those of “a recent study isolated two analogues from a Cephalothrix species collected from the Sea of Japan [41]” , did not identify these compounds using authentic standard of 1-hydroxy-4,4a-anhydro-8-epi-5,6,11-trideoxyTTX which was characterized by NMR.

The fragment ions at 162.07 as well as 110.07 are commonly shown for many TTX analogues. These fragment ions cannot exclude the possibility that it is 4,9-anhydro or 4,4a-anhydrodideoxyTTX. 8-epi-Type TTX analogues have been only reported in newts. That makes 4,9-anhydro or 4,4a-anhydrodideoxyTTX more plausible for identification of this peak than 1-hydroxy-4,4a-anhydro-8-epi-5,6,11-trideoxyTTX. I would recommend that you should mention the possibility that this peak can be also assigned as 4,9-anhydro or 4,4a-anhydrodideoxyTTX if the peak is not assigned with authentic standard in discussion.

Author Response

We have taken the referee's opinion on board and changed Compounds 6 and 10 to "unknown compounds #2 and #1 respectively in Table 1.

Changed the discussion (lines 221-31, 250, 317; Word file) to reflect that the identification of these compounds is putative/preliminary.

We have also added a sentence in the conclusions (lines 472-4; Word file)

Reviewer 2 Report

Please, some slight correction should be include in the manuscript. Chemistry format slight details.

Round 3

Reviewer 1 Report

I admit that your revision this time has improved the manuscript. However, I would still recommend that the structure of 1-hydroxy-4,4a-anhydro-8-epi-5,6,11-trideoxyTTX (10) should be removed. I think that this structure makes confusion. 

Author Response

Thank you for your suggestion. However, we don't feel the inclusion of structure 10 leads to any confusion. Removal of this structure would make the discussion on lines 221-231 confusing and more difficult to follow. As the inclusion of the structure on page 2 does no harm and provides for a more complete picture of the types of TTX analogues that are known, we would like to retain the structure.